# The Mechanisms of Benefit of High-Flow Nasal Therapy in Stable COPD

**DOI:** 10.3390/jcm9123832

**Published:** 2020-11-26

**Authors:** Massa Zantah, Aloknath Pandya, Michael R. Jacobs, Gerard J. Criner

**Affiliations:** Department of Thoracic Medicine and Surgery, Lewis Katz School of Medicine, Temple University, Philadelphia, PA 19140, USA; apandya1@gmail.com (A.P.); Michael.Jacobs@tuhs.temple.edu (M.R.J.); gerard.criner@tuhs.temple.edu (G.J.C.)

**Keywords:** COPD, chronic obstructive pulmonary disease, high flow nasal cannula, HFNC

## Abstract

High-flow nasal therapy (HFNT) is a unique system that delivers humidified, heated oxygen-enriched air via nasal cannula at high flow rates. It is a promising therapy for chronic obstructive pulmonary disease (COPD) patients. Several studies have examined the physiologic effects of this therapy in the patient population and have revealed that it improves mucociliary clearance, reduces nasopharyngeal dead space, and subsequently increases CO_2_ washout. It also improves alveolar recruitment and gas exchange. These mechanisms may explain the promising results observed in recently published studies that examined the role of HFNT in stable COPD patients.

## 1. Introduction

Chronic obstructive pulmonary disease (COPD) is one of the leading causes of mortality in the United States and worldwide. It is also considered a leading cause of disability, imposing an enormous burden on the US healthcare system. COPD prevalence, morbidity, and mortality vary across countries and across different groups within countries. The prevalence and burden are projected to increase over the coming decades due to continued exposure to COPD risk factors and aging of the world’s population. Unsurprisingly, the cost of care for COPD patients is strongly correlated with disease severity [1,2].

The disease is characterized by ventilatory limitation, leading to symptoms of airflow obstruction, including dyspnea, coughing, and excessive sputum production. As a result of this disease, patients may suffer from chronic hypoxemic and hypercapnic respiratory failure, or both, and are at risk of acute exacerbation events (AE) [3,4]. Current COPD pharmacotherapy includes bronchodilators, antibiotics, and corticosteroids. Despite the aggressive use of these agents in advanced COPD, many patients still fail to achieve adequate symptom control, improve airflow limitations, or reduce the risk of exacerbation. This has prompted a search for new treatment modalities that may help improve patients’ quality of life and reduce COPD morbidity and mortality.

Both long-term oxygen therapy (LTOT) and noninvasive ventilation (NIV) have been studied and are indicated in certain subgroups of COPD patients. Issues related to side effects and device compliance may limit the use of these interventions [5,6,7,8,9].

High-flow nasal therapy (HFNT) is a non-invasive respiratory support that supplies heated, humidified, oxygen-enriched air through a nasal cannula at high flow rates (up to 60 L per minute) [10]. Multiple studies have investigated the physiologic mechanisms that may explain some of the potential benefits of this modality. Some of the effects include the improvement of lung mucociliary clearance, flow-induced increase in airway pressure, the washout of upper-airway dead space, and reduction in dead-space ventilation [11,12]. In recent years, HFNT has gained popularity among intensivists treating patients with acute respiratory failure. To date, studies evaluating HFNT in COPD patients have mainly focused on those suffering from severe disease and those who require supplemental oxygen therapy [13,14,15]. Few studies have evaluated the effect of this modality on chronic hypercapnia [16,17]. In this article, we review the available evidence for using HFNT in stable COPD patients and focus on the mechanisms that may explain some of the promising results.

## 2. Nasal High-Flow Device and Gas Delivery

The nasal high-flow device is a humidifier with integrated flow generator that delivers high-flow warmed and humidified respiratory gases to patients through a variety of patient interfaces. It is intended for the treatment of spontaneously breathing patients who would benefit from receiving high-flow warmed and humidified respiratory gases. The device can be used on hospitalized patients with the intent of reducing the need for mechanical ventilation or on patients at home or in long-term care facilities. Flow rates may be adjusted from 2 to 60 L/min depending on the patient interface. There are a few expected side effects of using the device—if the humidification is too low, the subject may experience nasal and throat discomfort or epistaxis.

While other interfaces can be used, HFNT is frequently delivered using a nasal cannula. Cannulae come in different sizes, and one is selected based on patient comfort. There are a few side effects of using the nasal cannula, which are generally limited to some minor irritation or sensitivity where the cannula sits on the skin [10].

## 3. Physiologic Mechanisms of Benefits of HFNT in COPD Patients

HFNT has pleiotropic effects that improve work of breathing and oxygenation in those with respiratory failure. These include nasopharyngeal dead-space clearance, alveolar recruitment, improved gas exchange, mucociliary clearance, and improvement in dyspnea. Table 1 summarizes the physiologic mechanisms and the resulting clinical benefits of HFNT in stable COPD. 

### 3.1. Dead Space and CO_2_ Washout

Physiologic dead space in a normally functioning lung is near 30% and this can rise substantially in various disease states [18]. HFNT results in nasopharyngeal dead-space clearance with a “washout” of expired air in the upper airways [19]. Clearing CO_2_ from that dead space can improve ventilatory efficiency. Studies using inhaled inert gas show rapid clearance of the upper airway in a dose-dependent manner with increasing high flow rates. This has been demonstrated in both airway models and in test subjects [20,21]. Interestingly, the effect of HFNT on CO_2_ washout may extend to the trachea and lower airways and is also flow dependent. This was demonstrated in an experiment by Bräunlich et al. on an animal lung model [22]. This is an important mechanism that may explain the potential therapeutic effect of HFNT in treating hypercapnia. In fact, when compared to NIV, a study found that for HFNT use in stable hypercapnic COPD patients, there were no significant differences achieved in PaCO_2_ levels after treatment with either device [17]. Furthermore, the reductions in dead-space ventilation with using nasal high flow was found to be dependent on the physiologic, rather than the anatomic, dead-space volume. Biselli et al. conducted a study using HFNT on healthy controls and subjects with COPD during sleep (flow of 20 L/min and for intermittent periods of 5–10 min), using transcutaneous carbon dioxide monitoring under a metabolic hood. The study found similar responses to HFNT in both groups, with a significant decrease in dead-space ventilation (2.5 ± 0.4 L/min to 1.6 ± 0.4 L/min, *p* < 0.05) without a change in alveolar ventilation. The reduction in dead-space ventilation correlated with baseline physiologic dead-space fraction (r^2^ = 0.36, *p* < 0.05) but not respiratory rate or anatomic dead space volume [23]. This may explain why patients with COPD variable responses to HFNT may have, as it may be related to the degree of air trapping, hyperinflation, and subsequent physiologic dead space.

The above effects of HFNT on stable COPD patients during sleep were found to be superior to those of low-flow nocturnal oxygen. When compared to low-flow oxygen (2 L/min), room air delivered by HFNT (20 L/min) produced a greater reduction in tidal volume, and as a result, minute ventilation and work of breathing were both reduced and transcutaneous CO_2_ levels were lowered [24]. As expected, oxygen saturation improved with low-flow oxygen, but there was a clinically insignificant decrease in oxygen saturation with room air delivered by HFNT.

### 3.2. The Effect of Positive End-Expiratory Pressure

Alveolar recruitment can also be achieved with HFNT due to the positive end-expiratory pressure (PEEP) effect at higher flow rates. Spontaneously breathing volunteers with closed mouths wearing a HFNT device are reported to have higher tidal volumes [25]. The amount of PEEP measured is proportional to the flow rate with an increase of PEEP of nearly 0.7 cm H_2_O with every 10 L/min increase in flow rate [26,27]. As flow rates reach 60 L/min, the PEEP can be close to 5 cm H_2_O when measured by esophageal manometry [28]. This translates to increased end-expiratory lung volumes when measured by electrical impedance [29].

### 3.3. Attenuation of Inspiratory Effort

Another important physiologic effect of HFNT on patients with COPD is the reduction in inspiratory effort. This effect is flow dependent and is comparable to NIV. A study by Rittayamai et al. examined using HFNT at flow rates increasing from 10 to 50 L/min on 12 hypercapnic COPD patients in mild to moderate exacerbation and after initial stabilization on NIV. The study found that using HFNT at 30 L/min significantly reduced inspiratory effort (as estimated by simplified esophageal pressure–time product, sPTP_es_), similar to NIV delivered at modest levels of pressure support [30]. The reduction in inspiratory effort translates into decreased work of breathing. Furthermore, a physiologic study conducted by Mussi et al. on patients with acute exacerbation of COPD found that the use of HFNT led to a significant reduction in the neuroventilatory drive (as measured by electrical diaphragmatic activity) and work of breathing compared to conventional oxygen therapy (COT) [31]. Similarly, a study by Longhini et al. found that HFNT maintains the diaphragmatic displacement and improves patient comfort during periods of NIV interruption compared to COT in COPD patients with acute hypercapnic respiratory failure [32].

### 3.4. Potential Effects on Gas Exchange

Improved alveolar ventilation as a part of minute ventilation and alveolar recruitment from PEEP effects are likely major factors in the improvement in gas exchange noted with HFNT use. In patients immediately post extubation, HFNT was associated with improved PaO_2_/FiO_2_, comfort with oxygen delivery device, less oxygen desaturation and need for reintubation when compared to non-rebreathing oxygen masks [33,34]. Studies on patients presenting to an emergency department with acute hypoxic respiratory failure show that the use of HFNT was associated with higher rates of recovery from respiratory failure and significantly higher PaO_2_ levels, when compared to COT [35].

### 3.5. Improvement in Mucociliary Clearance

The flow rates of HFNT necessitate warming and humidification to prevent drying of mucous membranes. The efficiency of humidification is significantly increased at flow rates above 20 L/min and plateaus when reaching rates of 40 L/min or above [36]. For patient tolerability and for improved humidification, HFNT systems have their temperatures set to around 37 °C, close to the patient’s core temperature. The warm and humid HFNT air-gas mixture keeps mucous from becoming overly desiccated and improves mucociliary clearance [37].

### 3.6. Effects on Dyspnea and Work of Breathing

All of these effects contribute to the improvement in dyspnea and user comfort noted with HFNT use. Decreased accessory respiratory muscle use and respiratory rate is seen in HFNT compared to COT [38,39]. In patients with acute hypoxic respiratory failure, HFNT was associated with significant improvement in dyspnea and less discomfort when compared to non-invasive ventilation [40]. HFNT is also associated with improved exercise tolerance and rapid shallow breathing index compared to COT [41].

## 4. Current Clinical Evidence of HFNT Use in Stable COPD Patients

Several studies have examined the use of HFNT in stable COPD patients. A study by Storgaard et al. randomized 200 COPD patients with chronic hypoxemic respiratory failure who were already on LTOT (mean flow rate 1.6 L/min) to receive either HFNT in addition to LTOT or LTOT alone for a year. Although the study found a decrease in patient-reported acute exacerbation rates in the HFNT group, there was no difference in all-cause mortality or hospital admissions between the two groups. However, dyspnea symptoms, quality of life, PaCO_2_, and exercise capacity all significantly improved in the HFNT group compared to the LTOT group. Patients were instructed to use HFNT for 8 h per day, preferably at night. The mean adherence to HFNT was only 6 h per day and the flow rate was 20 L/min (the highest flow rate available on the device at the time of this study). This study highlights the potential role of HFNT as an adjunct to LTOT in COPD patients with chronic hypoxemic respiratory failure [15].

Another group of COPD patients where the use of HFNT has been evaluated is the chronic hypercapnic respiratory failure group. A study by Bräunlich et al. compared HFNT to NIV in stable COPD patients with hypercapnia (mean PaCO_2_ 56.5 mmHg). This crossover study randomized 102 patients, comparing 6 weeks of HFNT followed by 6 weeks of NIV and vice versa with the primary outcome being change in pCO_2_ level. CO_2_ levels decreased by 4.7% (95% CI 1.8–7.5, *p* = 0.002) using NHF and by 7.1% (95% CI 4.1–10.1, *p* < 0.001) from baseline using NIV. The study confirmed the noninferiority of HFNT compared to NIV in regard to reduction of PCO_2_, and thus NHF can be used as an alternative to NIV in this patient population. The mean duration of HFNT usage was significantly longer (5.2 ± 3.3 h/day compared to 3.9 ± 2.5 h/day in the NIV group), which may be due to easier usage and better tolerance of HFNT compared to NIV. In addition, the flow rate in the HFNT group was limited to 20 L/min (again, the highest flow rate available on the device at the time of the study’s inception) which may have limited the capacity of CO_2_ washout, because it is known that this physiologic effect is flow dependent [16].

When compared to standard of care, a study by Rea et al. found that using HFNT for one year resulted in fewer exacerbation days (18.2 versus 33.5 days; *p* = 0.045), increased time to first exacerbation (52 versus 27 days; *p* = 0.0495), and reduced exacerbation rate (2.97/patient/year versus 3.63/patient/year; *p* = 0.067). However, the study included mixed groups of patients with COPD and bronchiectasis, or both, and the mean use of HFNT was only 1.6 h per day. This was an issue with the study design, rather than tolerability, as the patients were instructed to use the equipment for only 2 h per day. The majority of patients in the HFNT group wished to continue using the device after study completion [42].

Finally, evidence from small studies showed that using HFNT during exercise testing in stable COPD patients leads to an increase in exercise duration, tolerance, and improvement in respiratory mechanics compared to standard oxygen therapy. This may be explained by the mechanism of dead-space elimination and the increase of CO_2_ washout, which may help reduce dynamic hyperinflation, a main exercise limitation in COPD patients. Table 2 summarizes some of benefits and drawbacks of HFNT.

## 5. Conclusions and Future Directions

The physiologic mechanisms of the benefits of HFNT may explain some of the promising results in the recent literature. Based on the available evidence, future studies should further investigate the role of HFNT in stable COPD patients to definitively determine the effect of this therapy on hospitalization and exacerbation rates. This therapy may be particularly beneficial as an add-on therapy to LTOT in COPD patients with chronic hypoxemic respiratory failure, more specifically patients with frequent exacerbations. In addition, it is important to investigate the role of HFNT in stable hypercapnic COPD patients as an alternative therapy to NIV, by using higher flows rates now that these devices are available. This is based on the fact that CO_2_ washout is flow dependent and using higher flows could lead to further increase in CO_2_ washout. Finally, the role of HFNT during pulmonary rehabilitation and exercise programs should be investigated. This would support expanded uses based on the promising results that have been published on HFNT use during exercise testing.

## Figures and Tables

**Table 1 jcm-09-03832-t001:** Summary of the physiologic mechanisms and the resulting clinical benefits of high-flow nasal therapy (HFNT) in stable chronic obstructive pulmonary disease (COPD) patients.

Physiologic Mechanism	Clinical Benefit
Nasopharyngeal dead-space clearance	Reduction in dead-space ventilation and possible improvement of hyperinflation
Decrease in minute ventilation	Improvement in work of breathing
Provides positive end-expiratory pressure	Alveolar recruitment and improvement in gas exchange
Reduction in inspiratory effort	Improvement in work of breathing
Delivery of warm and humidified oxygen	Improvement in mucociliary clearance

**Table 2 jcm-09-03832-t002:** Benefits and drawbacks of HFNT.

Benefits	Drawbacks
Improvement in dyspnea	Intolerance of noise, especially with higher flow
Dead space clearance	Abdominal distension
Can titrate FiO_2_	Avoid use in facial trauma/surgery
Improved work of breathing	Decreased mobility
Improved gas exchange	Increased risk of aspiration
Delivery of warm and humidified oxygen	May cause epistaxis
Can be used at home	Can cause nasal discomfort

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
