# Peer review of "The Mechanisms of Benefit of High-Flow Nasal Therapy in Stable COPD"

_jcm, 2020, doi:10.3390/jcm9123832_

Round 1
Reviewer 1 Report
This is an interesting well written manuscript focused in a relevant topic. I have only some minor suggestions.
I think in the introduction more information on the magnitude of COPD would be interesting. Also some more information on the use of High-Flow Nasal Therapy in COPD patients in US or other countries would be useful.
The paragraph “HFNT supplies heated, humidified, oxygen-enriched air through a nasal cannula at high flowrates (up to 60 liters per minute). Multiple studies have investigated the physiological mechanisms that may explain some of the potential benefits of this modality. Some of effects include the improvement of lung mucociliary clearance, flow induced increase in airway pressure, the washout of upper airway dead space, and the generation of a low level of positive airway pressure.” Needs a reference.
Author Response
We thank the reviewer for the time and effort in going over our paper.
In regards to your first comment, we have expanded on the introduction to explain the magnitude, and burden of the disease. In addition, we have mentioned how HFNT has gained the popularity among the intensivists to be used for acute hypoxic respiratory failure, and later for COPD. We also added citations to the last paragraph as you recommended:
"Chronic obstructive pulmonary disease (COPD) is one of leading causes of mortality in the United States and worldwide. It is also considered a leading cause of disability, imposing an enormous burden on the US healthcare system. COPD prevalence, morbidity and mortality vary across the countries and across different groups within countries. The prevalence and burden are projected to increase over the coming decades due to continued exposure to COPD risk factors and aging of the world’s population. Not surprisingly, the cost of care to patients with COPD is strongly related to disease severity.1,2
The disease is characterized by ventilatory limitation leading to symptoms of airflow obstruction including dyspnea, cough and excessive sputum production. As a result of this disease, patients may suffer from chronic hypoxemic and/or hypercapnic respiratory failure and are at risk for acute exacerbation events (AE). 3,4 Current COPD pharmacotherapy includes bronchodilators, antibiotics, and corticosteroids. Despite the aggressive use of these agents in advanced COPD, many patients still fail to achieve adequate symptom control, improve airflow limitations, or reduce the risk of exacerbation. This has prompted a search for new treatment modalities that may help improve patients’ quality of life and reduce COPD morbidity and mortality.
Both long-term oxygen therapy (LTOT) and noninvasive ventilation (NIV) have been studied and are indicated in certain subgroups of COPD patients. Issues related to side effects and device compliance may limit the use of these interventions.5–9
HFNT is a non-invasive respiratory support that supplies heated, humidified, oxygen-enriched air through a nasal cannula at high flow rates (up to 60 liters per minute).10 Multiple studies have investigated the physiological mechanisms that may explain some of the potential benefits of this modality. Some of the effects include the improvement of lung mucociliary clearance, flow induced increase in airway pressure, the washout of upper airway dead space, and reduction in dead space ventilation.11,12 In the recent years, HFNT has gained an important popularity among intensivists to manage patients with acute respiratory failure. To date, studies on COPD patients have mainly focused on severe disease and those who require supplemental oxygen therapy.13–15 Few studies evaluated the effect of this modality on chronic hypercapnia.16,17 In this article, we review the available evidence for using HFNT in stable COPD patients and focus on the mechanisms that may explain some of the promising results."
Reviewer 2 Report
Zantah et al reviewed the literature about high flow oxygen through nasal cannula in stable COPD patients.
Bibliography seems to be quite complete and updated. I have only few minor comments.
1) Heading “Physiologic mechanisms of benefits of HFNT in COPD patients”. I would divide the tet into some subheadings, such as pharyngeal dead space wash out, positive end-expiratory pressure, and so on.
2) Please note that in respiratory effort reduction, authors refer to a study by Rittayamai et al conducted in exacerbated COPD patients. There are at least 2 more studies regarding this issue (PMID 30882477 and 30071876)
Author Response
We thank the reviewer for the thoughtful suggestions.
- We have divided the paragraph explaining the physiologic benefits of the HFNT in stable COPD patients into the following sub headings:
Dead space and CO2 washout
The effect of positive end-expiratory pressure
Attenuation of inspiratory effort
Potential effects on gas exchange
Improvement in mucociliary clearance
Effects on dyspnea and work of breathing
- We also thank you for pointing out to the studies by Di Mussi and Longhini that looked at reduction in inspiratory effort in patients with COPD treated with HFNT. We have expanded on this paragraph to include these two studies.
" Attenuation of inspiratory effort
Another important physiologic effect of HFNT on patients with COPD is the reduction in inspiratory effort. This effect is flow dependent and is comparable to NIV. A study by Rittayamai et al. examined using HFNT at flow rates increasing from 10 to 50 L/min on 12 hypercapnic COPD patients in mild to moderate exacerbation, and after initial stabilization on NIV. The study found that using HFNT at 30 L/min significantly reduced inspiratory effort (estimated by simplified esophageal pressure–time product, sPTPes), similar to NIV delivered at modest levels of pressure support.29The reduction in inspiratory effort translates into decrease work of breathing. Furthermore, a physiologic study conducted by Di Mussi and colleagues on patients with acute exacerbation of COPD, found that the use of HFNT led to significant reduction in the neuroventilatory drive (measured by electrical diaphragmatic activity) and work of breathing compared to conventional oxygen therapy (COT).30 Similarly, a study by Longhini et al. found that HFNT maintains the diaphragmatic displacement and improves patient comfort during periods of NIV interruption compared to COT in COPD patients with acute hypercapnic respiratory failure .31"
Reviewer 3 Report
Thanks for the opportunity to review this paper. Its well written und gives compact informations about the field of NHF in stable COPD.
Some recommendations/ questions I want to address:
L35/36 Two times increase or positive airway pressure mentioned; I would add the reduction in dead space ventilation.
L36 The idea of wash-out upper airways was conducted by the paper of Möller et al. PMID25882385. The experimental wash-out was done during stops of breathing. Thats why only nasopharyngeal dead space wash-out was measured. In an experimental study by Bräunlich et al. CO2-wash-out was also measured in deeper airways PMID28396200 with significant decrease without any ventilation.
L54 In the study by Biselli no increase in alveolar ventilation was detected.
L57 Please cite!
L59 Do you mean rebreathing of CO2-rich air from airways. I think rebreathing of exhaled air is difficult. In general I feel this sentence is misleading.
Author Response
We thank you the reviewer for the thorough review and the valuable comments.
- L35/36 Two times increase or positive airway pressure mentioned; I would add the reduction in dead space ventilation: we made some changes on this paragraph to add your point: "Some of the effects include the improvement of lung mucociliary clearance, flow induced increase in airway pressure, the washout of upper airway dead space, and reduction in dead space ventilation. 11,12"
- L36 The idea of wash-out upper airways was conducted by the paper of Möller et al. PMID25882385. The experimental wash-out was done during stops of breathing. Thats why only nasopharyngeal dead space wash-out was measured. In an experimental study by Bräunlich et al. CO2-wash-out was also measured in deeper airways PMID28396200 with significant decrease without any ventilation: We thank the reviewer for bringing an interesting point. We have expanded on this paragraph and brought up this interesting observation: "Interestingly, the effect of HFNT on CO2washout may extend to the trachea and lower airways and is also flow-dependent. This was demonstrated in an experiment by Bräunlich, J and colleagues on an animal lung model.21
This is important mechanism that may explain the potential therapeutic effect of HFNT in treating hypercapnia" - L54 In the study by Biselli no increase in alveolar ventilation was detected: We appreciate your thorough review of this paper. That is accurate. We added this finding to our paper. "Biselli and colleagues performed a study using HFNT on healthy controls and subjects with COPD during sleep (flow of 20 L/min and for intermittent periods of 5-10 minutes), using a transcutaneous carbon dioxide monitoring under a metabolic hood. The study found similar response to HFNT in both groups, with a significant decrease in dead space ventilation (2.5±0.4 L/min to 1.6±0.4 L/min, p<0.05) without a change in alveolar ventilation."
-
L57 Please cite! We added more references to this paragraph: "Physiologic dead space in a normally functioning lung nears 30% and this can rise substantially in various disease states.18 HFNT results in nasopharyngeal dead space clearance with “wash out” of expired air in the upper airways.19 Clearing carbon dioxide from that dead space can improve ventilatory efficiency. Studies using inhaled inert gas show rapid clearance of the upper airway in a dose dependent manner with increasing high flow rates. This has been demonstrated in both airway models and in test subjects.20,21"
The references are :
18. Robertson, H. T. Dead space: the physiology of wasted ventilation. Eur Respir J 45, 1704–1716 (2015).
19. Frizzola, M. et al. High‐flow nasal cannula: Impact on oxygenation and ventilation in an acute lung injury model. Pediatr Pulm 46, 67–74 (2011).
20. Möller, W. et al. Nasal high flow clears anatomical dead space in upper airway models. J Appl Physiol 118, 1525–1532 (2015).
21. Möller, W. et al. Nasal high flow reduces dead space. J Appl Physiol 122, 191–197 (2017).
-
L59 Do you mean rebreathing of CO2-rich air from airways. I think rebreathing of exhaled air is difficult. In general I feel this sentence is misleading. We thank the reviewer for pointing this out and we realize that this phrase can be misleading. We do mean rebreathing CO2-rich air from the airways rather than exhaled air. To avoid the confusion, we have decided to eliminate this sentence.